# Valproic Acid Treatment after Traumatic Brain Injury in Mice Alleviates Neuronal Death and Inflammation in Association with Increased Plasma Lysophosphatidylcholines

**DOI:** 10.3390/cells13090734

**Published:** 2024-04-23

**Authors:** Regina Hummel, Erika Dorochow, Sonja Zander, Katharina Ritter, Lisa Hahnefeld, Robert Gurke, Irmgard Tegeder, Michael K. E. Schäfer

**Affiliations:** 1Department of Anesthesiology, University Medical Center Johannes Gutenberg-University Mainz, 55131 Mainz, Germany; rehummel@uni-mainz.de (R.H.); katharina.ritter@unimedizin-mainz.de (K.R.); 2Institute of Clinical Pharmacology, Medical Faculty, Goethe-University Frankfurt, 60596 Frankfurt am Main, Germany; dorochow@med.uni-frankfurt.de (E.D.); hahnefeld@med.uni-frankfurt.de (L.H.); gurke@med.uni-frankfurt.de (R.G.); 3Fraunhofer Institute for Translational Medicine and Pharmacology (ITMP), Fraunhofer Cluster of Excellence for Immune-Mediated Diseases, Theodor-Stern-Kai 7, 60596 Frankfurt am Main, Germany; 4Focus Program Translational Neurosciences (FTN), Johannes Gutenberg-University Mainz, 55131 Mainz, Germany; 5Research Center for Immunotherapy (FZI), Johannes Gutenberg-University Mainz, 55131 Mainz, Germany

**Keywords:** valproic acid, traumatic brain injury, neuroinflammation, lipidomic and metabolomic analyses, lysophoshatidylcholine, threonic acid

## Abstract

The histone deacetylase inhibitor (HDACi) valproic acid (VPA) has neuroprotective and anti-inflammatory effects in experimental traumatic brain injury (TBI), which have been partially attributed to the epigenetic disinhibition of the transcription repressor RE1-Silencing Transcription Factor/Neuron-Restrictive Silencer Factor (REST/NRSF). Additionally, VPA changes post-traumatic brain injury (TBI) brain metabolism to create a neuroprotective environment. To address the interconnection of neuroprotection, metabolism, inflammation and REST/NRSF after TBI, we subjected C57BL/6N mice to experimental TBI and intraperitoneal VPA administration or vehicle solution at 15 min, 1, 2, and 3 days post-injury (dpi). At 7 dpi, TBI-induced an up-regulation of REST/NRSF gene expression and HDACi function of VPA on histone H3 acetylation were confirmed. Neurological deficits, brain lesion size, blood–brain barrier permeability, or astrogliosis were not affected, and REST/NRSF target genes were only marginally influenced by VPA. However, VPA attenuated structural damage in the hippocampus, microgliosis and expression of the pro-inflammatory marker genes. Analyses of plasma lipidomic and polar metabolomic patterns revealed that VPA treatment increased lysophosphatidylcholines (LPCs), which were inversely associated with interleukin 1 beta (*Il1b*) and tumor necrosis factor (*Tnf*) gene expression in the brain. The results show that VPA has mild neuroprotective and anti-inflammatory effects likely originating from favorable systemic metabolic changes resulting in increased plasma LPCs that are known to be actively taken up by the brain and function as carriers for neuroprotective polyunsaturated fatty acids.

## 1. Introduction

Traumatic brain injury (TBI) is a major cause of death or long-term disability, and except for supportive care, therapeutic options are limited [1]. Multifactorial secondary processes are initiated directly after the injury. These include an impaired autoregulation of cerebral perfusion, disruption of the blood–brain barrier (BBB), edema formation, oxidative stress, and mitochondrial dysfunction, culminating in progressive neuronal cell death and brain inflammation [2,3,4]. Brain inflammation is associated both with short- and long-term consequences of TBI, such as post-traumatic epilepsy, neuropsychiatric disorders and dementia [1,5,6]. Up to date, clinical studies with hormone, antioxidant or immunosuppressive approaches including, e.g., treatments with progesterone, erythropoietin, cyclosporine or the interleukin 1 receptor antagonist anakinra after TBI were not or only moderately successful [7,8]. The association between inflammation and neurodegeneration in TBI is complex and includes the activation of brain-resident and peripheral cells infiltrating the damaged brain tissue, resulting in an amplification of the initial inflammatory response, which per se is not detrimental. The cells that contribute to the response, particularly microglia and astrocytes, are highly diverse, show remarkable plasticity, and serve important functions in healing processes including brain tissue clearance and scar formation as well as neuronal and synaptic recovery [9,10,11]. Therefore, near-complete suppression of acute inflammatory responses does not appear to be a promising strategy in TBI [12].

Studies in animal models of TBI have identified FDA-approved drugs for disorders other than TBI that demonstrate both neuroprotective and anti-inflammatory properties, making drug repurposing a promising option [13,14]. Valproic acid (VPA), an anticonvulsive drug used in epilepsy and mood disorders [15] as well as in post-traumatic seizure [16], was recently reported to attenuate neurological impairment, brain lesion size and adverse metabolic changes in a porcine model of TBI [17,18]. In rodent models of TBI, VPA improved neuronal survival, inhibited microglia activation and reduced brain lesion size [19,20,21]. Since VPA is a histone deacetylase inhibitor (HDACi), and histone acetylation promotes gene transcription via the transformation of tightly packed heterochromatin to loosely packed euchromatin [22], an epigenetic modulation of gene expression might be involved in these and other therapeutic effects of VPA [15,23]. HDACi function of VPA has been further implicated in differential activation and inhibition of transcription factors [24,25] including the transcriptional master suppressor REST/NRSF, which regulates neurogenesis, neuronal differentiation and neuroplasticity in the developing and adult brain via epigenetic remodeling [26,27].

However, the clinical use of VPA also carries risks including hepatotoxicity and pancreatitis [28] that have been attributed to alterations of phospholipid metabolism in the liver [29]. VPA-evoked abnormal liver function tests (alanine amino transferase, ALT) were associated with low levels of serum lysophosphatidylcholines (LPCs), ceramides and sphingomyelins, whereas triglycerides were markedly elevated and positively associated with abnormal ALT [29]. Another study reported multiple metabolic changes in association with VPA including plasma glucose, lactate, acetoacetate, VLDL/LDL, lysophosphatidylcholines (LPCs), phosphatidylcholines, choline, creatine, amino acids, N-acetyl glycoprotein, pyruvate and uric acid [30]. These findings suggest that VPA-evoked metabolic changes are complex—positive and negative depending on the source and context—and contribute in part to its effects on the brain. We have recently shown that abnormal behavior following TBI in mice is associated with long-lasting alterations of brain lipids, mainly low phosphatidylethanolamines and high triglycerides [31], while others showed lipidome changes within 24 h after TBI in rats [32,33]. In addition, clinical studies found differential phospholipid profiles in cerebrospinal fluid during the acute phase of TBI between surviving and deceased patients [34], and recently, a large human study revealed an inverse correlation between plasma levels of LPCs and TBI severity, suggesting that LPCs in this context provided protective effects or were a marker for favorable metabolic changes [35]. Inversely, TBI per se altered lipid metabolism and inflammatory markers in the liver [36], suggesting intricate brain-to-liver and liver-to-brain pro-inflammatory lipid signaling which might be targetable with VPA. Hence, lipid metabolism in the brain and periphery may be crucial for the outcome of TBI.

Here, we assessed the hypothesis that VPA has overall beneficial effects after experimental TBI, rescues gene repression mediated by REST/NRSF and switches metabolism to favorable lipid profiles. To this end, C57BL/6N mice were subjected to the controlled cortical impact (CCI) model of TBI and received VPA or vehicle 15 min, 1, 2, and 3 days post injury (dpi). Mice were monitored for neurological deficits and sacrificed at 7 dpi to examine brain pathology combining (immuno-) histology, immunoblot, gene expression analyses and plasma lipidomic and metabolomics analyses.

## 2. Materials and Methods

### 2.1. Mice

Animal studies are reported and conducted in compliance with the ARRIVE guidelines and the institutional guidelines of the Johannes Gutenberg University, Mainz, Germany and were approved by the animal Care and Ethics Committee of the Landesuntersuchungsamt Rheinland-Pfalz (Tierversuchsantrag protocol number 23177-07 G16-01-22). Male C57BL6/N mice (Janvier, ~22 g and 6 weeks old), a commonly used mouse strain in experimental research, were housed individually and maintained in a controlled environment (12 h dark/light cycle, 23 °C, 55% humidity) with food and water ad libitum. Only male mice were included to exclude fluctuations in the estrus-dependent hormonal status as possible confounding factors.

Mice were randomized into the following groups: CCI vehicle *n* = 12, CCI VPA *n* = 12, sham vehicle *n* = 8, sham-VPA *n* = 8. One animal of the CCI vehicle group died at 1 dpi after experimental TBI, and therefore, all subsequent analyses were performed on 11 animals in the CCI vehicle group. Group assignment was blinded to the experimenters who performed data acquisition and analyses. This experimental and exploratory animal study was not pre-registered. Animal numbers were determined by power analysis using experience from prior experiments [37]. The sample size of the CCI groups was based on the assumption that a 20% change in brain lesion size as the main outcome parameter is relevant. The probability of type-1 error was set to α = 0.05, and the probability of type-2 error was set at β = 0.2.

### 2.2. Drug Administration, Anesthesia, and Trauma Induction

VPA (VPA sodium salt, CAS Number 1069-66-5, Sigma-Aldrich, PubChem Substance ID 24277744) was dissolved in 0.9% NaCl and injected intraperitoneally (i.p., 400 mg/kg body weight). Vehicle animals received i.p. injections of 0.9% NaCl only. Injections were administered 15 min after the CCI/sham procedure and at 1, 2, and 3 dpi (Appendix A). Drug doses and the i.p. route of administration were based on previous work confirming adequate levels of VPA in the brain after i.p. injections [38].

CCI was performed during daytime essentially as described [39]. Perioperative body temperature as well as anesthesia and surgery duration were not different between groups (Appendix A). Briefly, animals were anesthetized with isoflurane inhalation (induction 4%, maintenance 2.1% (*v*/*v*)) to ensure sufficient anesthesia during the whole procedure. Rectal temperature was maintained at 37 °C with a feedback-controlled heating pad (Hugo Sachs, March Hugstetten, Germany). After midline incision and craniotomy, CCI was induced above the right parietal cortex with an electromagnetically controlled stereotaxic impactor (Leica BioSystems, Wetzlar, Germany, Impact One; velocity: 6 m/s; duration: 200 msec; impact depth: 1.5 mm). The post-operative survival time was 7 days. Sham mice were treated equally to CCI mice in terms of anesthesia, skin incision and wound closure but without craniotomy. Craniotomy itself can contribute to brain damage in the CCI model, and therefore, only light non-penetrating drilling on the bone surface was performed to allow comparisons between non-injured and injured brains [40]. Before and after TBI, the neurobehavioral outcome of mice was assessed one day before surgery and at 1, 3, 5 and 7 dpi using a neurological severity score (NSS) and the rotarod (RR) test essentially as described [39]. The NSS comprised 9 different tasks to evaluate the motor ability, alertness, balancing and general behavior (severe neurological dysfunction NSS = 12; see timeline). Three experimenters did the experiments, all in a blinded manner, one performing the sham or CCI procedure, one performing drug or vehicle application and the third performing behavioral analyses. All post-interventional analyses were performed in a blinded and unbiased fashion.

### 2.3. Lipidomic and Metabolomic Analyses of Plasma

At 7 dpi, mice were anesthetized and killed by decapitation. Blood was collected in a 50 mL falcon tube with 80 µL heparin and centrifuged (Centrifuge 5804 R, Eppendorf AG; Hamburg, Germany) at 3800 rpm for 8 min. The plasma (approximately 200 µL) was transferred into a 1.5 mL tube and stored at −80 °C until further use.

Lipidomic and metabolomic analysis were conducted applying the same procedure as previously described [41]. Further details are described in the Appendix A. Briefly, a methyl-tert-butyl-ether (MTBE)-based liquid–liquid extraction was used to allow for the simultaneous analysis of polar metabolites and lipids from the same sample. For lipid extraction, 75 µL of internal standards (IS) in methanol, 250 µL of MTBE and 50 µL of 50 mM ammonium formate were added to 10 µL of mouse plasma and mixed vigorously. After centrifugation (20,000× *g*, 5 min, ambient temperature), the upper phase was transferred to a new tube, and the lower phase was reextracted using 100 µL of a mixture of MTBE: methanol: water (10:3:2.5, *v*/*v*/*v*, upper phase). For the measurement of lipids, the combined upper phases were evaporated under a gentle stream of nitrogen at 45 °C, stored at <−70 °C. For the measurement of polar metabolites, 200 µL of acetonitrile was added to the lower phase and mixed by gentle shaking. The mixture was then dried under a nitrogen stream at 45 °C and reconstituted in 100 µL acetonitrile/water (50:50 *v*/*v*) before analysis. Prior to lipid analysis, samples were reconstituted in 100 µL of methanol.

Analyses of lipids and polar metabolites were performed on an Orbitrap Exploris 480 with a Vanquish horizon UHPLC system (both Thermo Fisher Scientific, Dreieich, Germany). Data were acquired using XCalibur v4.4, and relative quantification was performed in TraceFinder 5.1 (both Thermo Fisher Scientific, Dreieich, Germany).

For lipidomic analysis, a Zorbax RRHD Eclipse Plus C8 1.8 µm 50 × 2.1 mm ID column (Agilent, Waldbronn, Germany) with a pre-column of the same type was used, and a linear gradient was run over 14 min. The mobile phases were (A) 0.1% formic acid and 10 mM ammonium formate and (B) 0.1% formic acid in acetonitrile/isopropanol (2:3, *v*/*v*). Full-scan spectra were acquired from 180 to 1500 *m*/*z* at 120,000 mass resolution each 0.6 s and data-dependent MS/MS spectra at 15,000 mass resolution in between. The relative quantification of previously identified lipids was performed in TraceFinder 5.1 using a mass error of ±3 ppm for positive and ±5 ppm ionization mode, and identification was further supported by the isotope ratio and the comparison of the MS/MS spectra. Lipid data were calculated as area ratios to one internal standard per lipid class.

For metabolomic analysis, polar metabolites were separated on a SeQuant ZIC-HILIC, 3.5 µm, 100 mm × 2.1 mm I.D. column coupled to a guard column with the same chemistry (both Merck, Darmstadt, Germany) and a KrudKatcher inline filter (Phenomenex, Aschaffenburg, Germany). Using 0.1% formic acid in water (solvent A) and 0.1% formic acid in acetonitrile (solvent B), binary gradient elution was performed with a run time of 25 min. Full-scan spectra were acquired with a resolution of 120,000 at a mass range of 70–700 *m*/*z* and 59–590 *m*/*z* in positive and negative ionization mode, respectively. The acquisition of MS/MS spectra was performed in a data-dependent manner (ddMS^2^) at a resolution of 15,000. Analyzed polar metabolites were identified with a mass error of ±5 ppm by the assessment of the isotope ratio and by matching acquired MS/MS spectra to library spectra. Putative signals of VPA and its metabolites were identified in full-scan mode with a mass error of ±5 ppm and by assessing the corresponding isotope ratios. Relative quantification was performed based on peak areas in extracted ion chromatograms, which were normalized by median-based probabilistic quotient normalization (PQN) [42].

### 2.4. Immunofluoresence Analyses and Histology

Mice were anesthetized with 4 vol% isoflurane and then decapitated. Afterwards, their brains were dissected, immediately frozen in powdered dry ice and stored at −20 °C. Cryostat brain sectioning (HM 560 Cryo-Stat, Thermo Scientific, Dreieich, Germany) and cresyl violet staining were carried out as described [37]. Briefly, brains were cut to 12 µm thick sections with a 500 µm interval from Bregma +3.14 mm to Bregma −4.36 mm and collected on glass slides (Superfrost plus, Thermo Scientific, Dreieich, Germany). The quantification of brain lesion volumes in cresyl violet-stained sections was performed as described [43]. Digitalization and analysis were conducted in a blinded fashion using a bright field microscope (Stemi 305, Carl Zeiss, Oberkochen, Germany) and Zen software (Carl Zeiss, Oberkochen, Germany, RRID SCR_013672). The thickness of the dentate gyrus granule cell layer (GCL) was determined in the septal hippocampus across the suprapyramidal blade in ipsi- and contralesional hemispheres and in samples from sham animals. Mean values were calculated from four values determined across the suprapyramidal blade from two different sections between Bregma −1.7 mm and Bregma −1.9 mm as described [39].

For fluorescent immunohistochemistry, 12 µm frozen sections were air-dried for 30 min, fixed with 4% paraformaldehyde (PFA) for 10 min and blocked in PBS, pH 7.4 containing 5% goat serum, 0.5% bovine serum albumin and 0.1% Triton X-100. The following primary and secondary antibodies were used (designation, dilution, supplier, RRID): rabbit anti-Iba1, 1:500, RRID:AB_839504; and rat anti-GFAP, 1:500, RRID:AB_86543, were diluted in blocking solution and incubated on sections over night at 4 °C; goat anti-rabbit Alexa Fluor 488, 1:500, RRID:AB_2576217 and goat anti-rat Alexa Fluor 568, 1:500, RRID:AB_141874) were incubated on sections at 22 °C for 1.5 h. All sections were counterstained with 4′,6-diamidino-2-phenyl-indol-dihydrochlorid (DAPI, 1:10.000, Sigma Aldrich, St. Louis, MO, USA) and mounted with Immu-Mount (Thermo Fisher, Dreieich, Germany). Immunohistochemical images were captured using a confocal laser-scanning microscope (LSM5 Exciter, Carl Zeiss, Oberkochen, Germany) using identical filter and acquisition parameters for all sections. The area percentage of anti-GFAP or anti-Iba1 immunoreactivity was determined in the perilesional brain cortex tissue between the midline and the lesion. Images were analyzed using ImageJ (NIH Image, RRID:SCR_003070) in a blinded and unbiased fashion. Images (8-bit) were processed using appropriate threshold adjustment and the “Analyze Particles” function essentially as described [44]. Data were expressed as the tissue area percentage occupied by anti-GFAP or anti-Iba1 immunostaining.

### 2.5. Gene Expression Analyses

Brain tissues from intermediate sections obtained during cryosectioning were cut along the midline to separate the ipsilesional and contralateral hemispheres; upper quadrants of the sections were collected, frozen with liquid nitrogen, stored at −80 °C and processed for RNA extraction and gene expression analyses as described [11]. Briefly, RNeasy and QuantiTect Reverse Transcription Kits (Qiagen, Hilden, Germany) were used to extract RNA and reverse transcribe to cDNA with gDNA removal. Analyses were conducted using a SYBRgreen Kit or Maxima Hot Start Kit (both Thermo Scientific) with primer and probes from Eurofins by quantitative polymerase chain reaction (qPCR) (Light Cycler 480, Roche, Basel, Switzerland). All values were normalized to the housekeeping gene cyclophilin A (*PPIA*), and absolute quantification was performed using gene-specific standard curves of mRNA copies [45]. Sequences of oligonucleotide primers pairs are available on request.

### 2.6. Identification of REST/NRSF Target Genes

We hypothesized that VPA effects in TBI are mediated at least in part via epigenetic regulation of the transcription factor REST/NRSF (pathway drawing Appendix A) that should manifest in expression changes of its target genes (Table 1). To address this hypothesis, we analyzed expressions of candidate REST/NRSF target genes. The candidates were identified/selected by searching promotor sequences for the 21 bp neuron restrictive silencer element (NRSE, also known as repressor element-1 (RE-1)) using e.g., https://www.genecards.org/, https://epd.epfl.ch//index.php, accessed on 1 October 2023). In addition, we searched gene ontology databases for genes with GO terms associated with REST/NRSF. Table 1 shows the results of our search and the chosen target genes which are grouped according to different biological processes.

### 2.7. Immunoblotting

Samples from ipsilesional brain tissue sections were collected during histological sectioning and processed for immunoblotting essentially as described [11]. Briefly, tissue sections were homogenized in ice-cold RIPA lysis buffer [composition: 50 mM Tris-HCl, pH 7.5, 150 mM NaCl, 1 mM EDTA, 1% (*v*/*v*) NP-40, 0.1% (*v*/*v*) sodium dodecyl sulfate (SDS), complete protease inhibitors (Roche, Basel, Switzerland)], protein concentrations were determined and equal amounts of proteins (30 µg per sample) were separated by SDS-PAGE and transferred to nitrocellulose membranes (PerBio Science). The following primary antibodies were used: rabbit anti-GFAP (1:1000, RRID:AB_10013382) and mouse anti-GAPDH (1:1000, RRID:AB_1616730) with appropriate secondary infrared dye-conjugated antibodies (1:15,000, goat anti-rabbit IgG IRDye 680, RRID:AB_621841 or goat anti-mouse IgG IRDye800CW, RRID:AB_10793856). To confirm HDACi activity of VPA, brain samples from VPA- or vehicle-treated CCI mice were pooled (*n* = 11, each group), loaded on two large lanes of the same SDS-Gel, transferred to a nitrocellulose membrane, and then membrane stripes were probed with antibodies of the Acetyl-Histone H3 Antibody Sampler Kit (#9927, Cell Signaling Technology, Danvers, MA, USA, RRID:AB_330200). To evaluate BBB integrity, proteins lysed in RIPA buffer (5 µg protein per dot) were spotted onto a nitrocellulose membrane, washed for 10 min in TBST and incubated with goat anti-mouse IRDye800 (1:10,000, RRID:AB_10793856) for one hour at 22 °C. Western blot protein bands or dot blot spot densities were revealed using the Odyssey SA Imaging System (LI-COR, Lincoln, NE, USA) and quantified with Image Studio (RRID:SCR_014579).

### 2.8. Statistical Analyses

Data were analyzed using GraphPad Prism^®^ software (RRID:SCR_002798). All data sets were assessed for normal distribution using the Shapiro–Wilk test and QQ plots; outliers were identified by an iterative Grubb’s test and excluded from further analysis as indicated in the figure legends. Parametric data were statistically analyzed using an unpaired Student’s *t*-test, and non-parametric data were analyzed using the Mann–Whitney *U*-test with two-sided tests. For multiple groups, one-way ANOVA or Kruskal–Wallis tests followed by post hoc analyses (Holm–Šidák or Dunn’s correction, respectively) were used as indicated in the figure legends. Statistical analyses including two factors, e.g., time and treatment for body weight and behavioral analyses were assessed using two-way ANOVA with α adjustment according to Šidák. Data are shown as mean ± SEM, data points correspond to individual animals, and sample sizes are given in figure legends. Tests were considered statistically significant at a multiplicity adjusted *p* value < 0.05. Significance levels are indicated by asterisks (* *p* < 0.05, ** *p* < 0.01, *** *p* < 0.001, **** *p* < 0.0001).

Data of lipidomic and metabolomics analyses were analyzed with GraphPad Prism 9.02, Origin Pro 2022 and MetaboAnalyst (https://www.metaboanalyst.ca/, accessed on 1 October 2023) [46]. Volcano plots were used to assess fold differences (x-axis) versus the negative Log10 of the *p*-values of group-vise *t*-tests (y-axis). Raw data (AUC of the analyte divided by the AUC of the internal standard, AUC/IS) were normalized to have a common mean and standard deviation of 1 (autoscaling in MetaboAnalyst) and were submitted to discriminant partial least square analysis (PLS-DA) to assess the differences between groups and find the analytes that contributed the most to variance between the groups. Variable importance plots (VIPs) show the top candidates. Linear regression analyses were used to assess the association of candidate lipids or polar metabolites with mRNA values (rt-PCR) of the key inflammatory markers, *Il1b* and *Tnfa*.

## 3. Results

### 3.1. VPA Functions as HDACi and REST/NRSF Is Up-Regulated after CCI

C57BL/6N mice were subjected to the CCI model of TBI and received VPA (400 mg/kg, i.p.) or vehicle treatment i.p. at 15 min, and 1, 2, and 3 dpi. Body weights at 7dpi were similar in the VPA and vehicle-treated CCI groups. To assess the HDACi activity of VPA at the given dose in the ipsilesional brain, samples from CCI animals were collected at 7 dpi and processed for immunoblotting using antibodies specific to different histone H3 acetylated lysine residues (Figure 1A). The results revealed an acetylation of histone H3 at Lys14 and Lys27 in vehicle-treated animals. VPA treatment increased the acetylation of histone H3 at Lys14 and Lys27, thereby confirming the expected HDACi function of VPA in the brain in our experimental setting.

Next, we analyzed the mRNA expression of the transcription factor REST/NRSF in brain samples dissected at different time points after CCI (Figure 1B, naive, 1, 3, 5, 7 dpi). Compared to samples from naïve, non-injured mice, REST/NRSF mRNA expression was about twofold increased at 3 dpi and remained up-regulated at this level until 7 dpi (Figure 1B).

### 3.2. VPA Treatment Does Not Affect mRNA Expression of REST/NRSF Target Genes in the Injured Brain

VPA inhibits the transcription-repressive functions of REST/NRSF in vitro [47], and we therefore expected that VPA administration increases the expression of REST/NRSF target genes. To test this hypothesis, we selected target genes that meet three criteria: (1) containing the 21 bp repressor element RE-1 [48], (2) being experimentally validated as REST/NRSF-regulated genes, and (3) having GO annotations as REST/NRSF regulation. We then assessed the expression of the top 15 candidates by qPCR analysis in ipsilesional brain samples of CCI/sham samples after VPA or vehicle treatment at 7 dpi (Figure 2). In agreement with our time-course analysis (Figure 1B), REST/NRSF mRNA expression was increased at 7 dpi compared to sham, but equally in both groups, i.e., without the effect of VPA (Figure 2A). Furthermore, the expression of 15 REST/NRSF target genes was also only slightly affected by VPA treatment, and none of these changes reached a statistically significant level (Figure 2B–P).

### 3.3. Administration of VPA Does Not Influence Acute Neurological Deficits or Brain Lesion Size 7 dpi but Attenuates Structural Damage in the Hippocampus

Compared to vehicle treatment, the relative body weight (expressed in percent of pre-OP body weight) of VPA-treated CCI and sham animals was reduced at 3 dpi and 5 dpi, but it was restored in CCI animals to the level of the vehicle group at 7 dpi (Figure 3A). Assessment of behavioral analyses using a neurological severity score (NSS) at 1, 3, 5, and 7 dpi revealed neurological deficits in the CCI group compared to sham animals regardless of VPA treatment (Figure 3B). Similarly, the rotarod performance to assess sensorimotor coordination was reduced after CCI but not different between the VPA and vehicle treatment groups (Appendix A).

At 7 dpi, mice brains were removed and processed for histopathological analyses. We examined the brain lesion size and structural integrity of the ipsilesional granule cell layer (GCL) of the hippocampus (Figure 3C–F). Brain lesion volumetry did not reveal differences between the VPA and vehicle treatment groups (Figure 3E). Determination of the GCL thickness, however, showed that VPA attenuated the structural damage of the GCL (Figure 3F). The results show that VPA did not affect acute neurological deficits or brain lesion size at 7 dpi but attenuated remote structural damage in the hippocampus.

### 3.4. VPA Treatment Does Not Affect BBB Disruption and Astrocyte Activation at 7 dpi

Since BBB dysfunction exacerbates secondary brain injury and VPA was reported to protect BBB integrity [49], we determined IgG brain extravasation as a proxy for BBB damage using an immuno-dot-blot assay at 7 dpi (Figure 4). Ipsilesional brain samples from CCI mice contained more IgG protein than those from sham mice, but IgG extravasation was not influenced by VPA treatment (Figure 4A). The activation of astrocytes is a hallmark of TBI, relevant for BBB function, and is a prominent histopathological feature in the CCI model of TBI [44,50]. We studied astroglial activation using GFAP as a marker in qPCR, immunoblot, and immunohistochemistry (Figure 4B–D). The experiments demonstrated CCI induced up-regulation of GFAP both at the mRNA (Figure 4B) and protein level (Figure 4C,D) equally in both treatment groups. A panel of exemplary immunofluorescence images is presented in Appendix A. There was no difference in GFAP expression between VPA and vehicle treatment in injured brains.

### 3.5. VPA Attenuates Microglia Activation and Pro-Inflammatory Gene Expression at 7 dpi

To investigate microglia activation, another hallmark of TBI, we determined the mRNA expression of marker genes by qPCR, which are known to be up-regulated by microglia after TBI (*Aif1*, *Tspo*, *Cd74)* (Figure 5A–C). As expected, all marker genes were up-regulated in ipsilesional brain samples from CCI mice irrespective of VPA treatment. However, VPA mildly attenuated the up-regulation of *Cd74* compared to vehicle treatment (Figure 5C). In addition, we assessed microglia via anti-Iba1 fluorescent immunohistochemistry in the perilesional cortex (Figure 5D–F). Iba1 immunoreactivity was attenuated in VPA-treated CCI mice as compared to vehicle-treated CCI mice (Figure 5F). A panel of additional exemplary immunofluorescence images is presented in Appendix A.

Furthermore, we assessed the gene expression of pro-inflammatory cytokines (*Il1b*, *Tnfa*, *Il6)* and inducible nitric oxide synthase (*Nos2*) (Figure 6A–D), which are produced by microglia and other cell types after TBI [11]. VPA treatment attenuated the expression of *Il1b*, and *Nos2* (Figure 6A,C) as compared to vehicle treatment, whereas *Tnfa* and *Il6* expressions were equally upregulated in the injured brain in both treatment groups (Figure 6B,D). Overall, VPA had significant but mild anti-inflammatory effects.

### 3.6. VPA Treatment Is Associated with High Levels of Lysophosphatidylcholines in Plasma

It has been recently shown that the outcome of TBI in patients depends on the profile of serum lipid species. In particular, patients with high levels of lysophosphatidylcholines (LPCs), ether-bound phosphatidylcholines (PC-Os) and sphingomyelins (SMs) had more favorable long-term outcomes than patients with low levels [35]. Motivated by this human study, we analyzed the lipidomic (Figure 7) and metabolomic profiles of polar metabolites (Figure 8) in plasma samples of VPA- and vehicle-treated CCI and sham mice at 7 dpi in association with the mRNA expression of the key pro-inflammatory markers IL-1β and TNFα.

Overall, 363 lipid species were reliably quantified in all treatment groups. Volcano plots comparing CCI-Vehicle versus CCI-VPA groups (Figure 7A) show an increase in LPC, PC and SM species in VPA-treated mice, i.e., of those lipid species which had been found to be associated with favorable outcomes in humans. Similar but even stronger differences were observed in Volcano plots by comparing sham-Vehicle versus sham-VPA mice (Appendix A).

Data obtained by LC-HRMS analysis revealed low remaining VPA concentrations in plasma at 7dpi, i.e., 4 days after cessation of treatment, which were higher but highly variable in sham-VPA mice than in CCI-VPA mice (Appendix A). Accordingly, treatment effects of VPA on lipids were somewhat stronger in sham-VPA mice than in CCI-VPA-mice.

Partial least square discrimination analysis (PLS-DA) shows overlapping scatter clouds (PLS-DA component 1 versus PLS-DA component 2 Figure 7B), but the variable importance plot (VPI, Figure 7C) agrees with the VPA-associated increase in LPC species, which were highest in sham-VPA animals and lowest in CCI-Vehicle mice. Importantly, levels of LPCs (sum of all LPC species) were negatively associated with the mRNA expression of IL-1β (Figure 7D) and TNFα (Appendix A); i.e., the higher LPC was, the lower was *Il1b* or *Tnfa*, suggesting that high LPC in plasma was associated with an attenuation of the pro-inflammatory transformation in the brain in response to the injury. *Il1b* was also negatively associated with phosphatidylcholines (sum of all PC species) and lysophosphatidylethanolamines (LPEs, sum of LPE species).

A detailed analysis of individual lipid species is presented in Appendix A. In addition to the differences in LPC levels (bottom panel in Appendix A), the results reveal an increase in LPE but decrease in PE in VPA-treated mice and high levels of long chain acyl-carnitines in CCI mice compared with sham irrespective of the treatment group (Appendix A). Furthermore, long-chain sphingomyelins were increased in VPA-treated mice (Appendix A), and triglycerides were higher in sham mice than CCI mice, which is explained by temporary reduction in food intake in CCI mice. It is of note that the plasma cholesteryl-ester (CE) values were somewhat higher in VPA-treated mice, which would point to an undesired metabolic effect.

### 3.7. VPA Treatment Is Associated with High Threonic Acid and Low Gentisic Acid

In addition to the observed high LPC serum levels in favorable-outcome TBI patients in the human metabolomic study mentioned above [35], there were some polar metabolites which differed between patients with favorable and unfavorable outcomes. One of these metabolites was threonic acid. Here, we could recapitulate this change in VPA-treated CCI mice. In addition to high threonic acid, we observed a reduction in N-acetylarginine and gentisic acid species (Figure 8A). The lowering of gentisic acid in VPA-treated mice was also evident in a Volcano plot comparison of sham-treated animals (Appendix A). PLS-DA separated CCI- from sham-treated mice (Figure 8B). CCI-VPA mice were more like sham mice than CCI-Vehicle mice, suggesting that VPA treatment partially restored metabolic profiles. The variable importance plot agrees with the results of the volcano plot comparisons (Figure 8C). XY-scatter plots showing the association of *Il1b* with key metabolites reveal that groups can be differentiated by *Il1b* versus threonic acid and by *Il1b* versus gentisic acid, suggesting that these were the most relevant metabolites (Figure 8D). The respective correlations analyses for *Tnfa* are shown in Appendix A.

## 4. Discussion

In this study, we report the repurposing of VPA, a clinically used drug for epilepsy and mood disorders, to treat mice after experimental TBI. We assessed the hypotheses that (a) VPA has neuroprotective and anti-inflammatory effects, (b) that these effects are mediated by the HDACi function of VPA, thereby inhibiting transcription-repressive actions of REST/NRSF, and (c) that VPA causes alterations in plasma lipid patterns.

Examination of brain lesion size did not reveal treatment effects of VPA. However, the structural damage of the hippocampal granule cell layer (GCL) was attenuated. Injury progression in the directly damaged cerebral cortex is a relatively rapid process that reaches a maximum at 3 days after injury [51,52] and precedes the degenerative sequelae in more distant regions such as the GCL in the CCI model of TBI [53]. Thus, GCL damage is a remote effect of the initial injury, and indirect protective effects of VPA via remote mechanisms may explain this observation.

VPA, considered an important regulator of innate and adaptive immune cells [54], was reported to stimulate autophagy and the Nrf2/ARE pathway after TBI, which resulted in a reduced activation of microglia [25]. Here, we examined the anti-inflammatory effects of VPA treatment, and our results agree with moderate anti-inflammatory effects, as evidenced by reduced microgliosis and reduced TBI-evoked up-regulation of the pro-inflammatory marker genes *Il1b*, *Cd74*, and *Nos2*. We did not find evidence for alterations in astrocyte reactivity or astrogliosis after VPA treatment. The observed “microglia-only” effect suggests a mild anti-inflammatory efficacy that does not spread from microglia to astrocytes. The latter receive activation signals from microglia via inflammatory cytokines [55]. Furthermore, neurological deficits, brain lesion size and BBB permeability were not attenuated by VPA treatment, again suggesting mild therapeutic efficacy within the 7 dpi time window.

VPA is a class I HDACi [23]. Dysbalanced histone acetylation is associated both with acute and chronic neurodegeneration [56,57], and augmentation of histone acetylation can reduce brain tissue damage and improve neurological outcome in experimental TBI [58]. We confirmed increased histone acetylation in brain lysates after CCI and VPA treatment and an up-regulation of REST/NRSF mRNA expression along with the CCI-induced reduction in some target genes [47,59]. However, REST/NRSF was up-regulated after TBI equally in both treatment groups. Consequently, the expression of REST/NRSF target genes was also similar in both treatment groups, and minor differences did not reach statistical significance, suggesting that the mild neuroprotective effects of VPA after TBI were not mediated via the de-repression of the REST/NRSF complex. Previous data indeed suggest that autophagy and the Nrf2/ARE pathway may be more relevant for VPA-mediated protection in this context [25], and further mechanisms of actions of VPA may contribute to the overall outcome including the potentiation of gamma-aminobutyric acid (GABA) signaling via the modulation of GABA metabolism [60] and activation of mTOR signaling [61].

Moreover, VPA was also reported to enhance phosphatidylinositol trisphosphate (PIP3) signaling and phosphoinositide turnover [62,63], suggesting the modulation of lipid signaling. VPA is known to alter inositol [64] and lipid metabolism [29], the latter mostly considered as unwanted metabolic side effects [65] or even as biomarkers of VPA-caused hepatotoxicity [29]. TBI per se may lead to liver inflammation [36], which may preclude VPA as a therapeutic option after TBI. However, VPA-mediated changes of sphingolipids and inositol were also considered to contribute to, or be associated with, the therapeutic effects in major depression [64] and refractory epilepsy [66], both presenting with abnormal lipidomic plasma profiles [66,67]. Particularly, VPA non-responders with refractory epilepsy had increased triglycerides and low glycerophospholipids [66]. These profile changes are in part reminiscent of previously observed perilesional lipid profile changes in mouse brains after CCI [31]. Hence, VPA’s metabolic impact appears to be inseparable from its wanted and unwanted effects. Therefore, we studied plasma lipidomic and metabolomic plasma profiles and found increased LPCs and threonic acid in VPA-treated mice—both CCI and sham groups, and reduced gentisic acid, again in both VPA groups. Hence, these are drug effects that do not require a CCI x VPA interaction. Indeed, the effects were somewhat stronger in sham-VPA mice, suggesting that CCI elicits opposing effects, which would agree with a recent human study showing that high plasma LPCs and threonic acid were associated with a favorable outcome after TBI [35]. Importantly, LPC plasma levels in our mice were inversely correlated with brain gene expression of the pro-inflammatory cytokines IL-1β and TNFα, showing that VPA induced changes in systemic lipid profiles related to anti-inflammatory effects in injured brains.

Further studies have revealed that alterations in lipid metabolism and particularly LPCs or a loss of LPCs play a role in the pathogenesis of post-TBI remodeling [68,69]. However, a high-dose local injection of LPC into the white matter is a common model to induce demyelination [70] and leads to a non-specific disruption of myelin lipids [70]. Importantly, increased endogenous LPC was not associated with brain damage [70], and there is no evidence that endogenous LPC is toxic to the brain. An in vitro study in BV-2 microglia found that the application of E. coli-derived LPC triggers inflammatory processes via caspase-mediated inflammasome activation [71], including the release of pro-inflammatory cytokines, which in turn stimulate a release of eicosanoids and reactive oxygen species that potentiate CNS injuries [72]. On the other hand, LPCs were also shown to promote PKA-induced phosphorylation of GSKβ in mycobacterium tuberculosis-infected bone marrow-derived macrophages, leading to the suppression of NF-κB activation and decreased secretion of pro-inflammatory cytokines, TNFα and IL-6. This study also provided evidence that LPCs promote phagosome maturation [73]. Moreover, LPCs were identified as one of the first “find-me“ signals that initiate the recruitment of phagocytes to the site of apoptosis [74]. Phagocytosis after TBI is crucial for the removal of dead cells, hematoma, and myelin debris, thereby facilitating the resolution of inflammation and neuronal repair [11,75]. Hence, obviously, the LPC sources and doses determine the outcome, and it must be considered that LPCs of different chain length and saturation may have differential effects. In our study, LPC species all tended to be increased in VPA groups, but the LPCs of 20 C-atoms were most strongly regulated [76].

There are limitations of this study that need to be considered. Clinical trials focusing on VPA to evaluate its safety and tolerability found a maximum tolerated dose of 140 mg/kg body weight in healthy volunteers [77]. In animal models, VPA exerts biological effects in the brain at higher concentrations (300–400 mg/kg) [19,25,78] without obvious liver toxicity. The serum protein binding of VPA differs substantially between humans (95%) and mice (12%), and protein binding is rate limiting for its clearance by the liver and therefore accounts for striking differences in the half-lives of the drug in different species [79]. VPA metabolism is much faster in mice than humans [80]. Considering the short half-life and low penetration of VPA to cross the BBB [81], we have chosen 400 mg/kg for repetitive administration after TBI (4 doses in total). An optimization of doses and schedules might have strengthened neuroprotective and anti-inflammatory effects, and lower doses might have prevented the temporary body weight loss during treatment. For example, biocompatible VPA-coupled nanoparticles [82] might improve brain uptake and efficacy and may be safe in acute brain injuries. Brain uptake, though, might not be essential because our lipidomic studies revealed major differences in plasma LPCs, which is an effect likely mediated through the effects of VPA in the liver. Hence, VPA may provide brain-protective lipid/metabolic switches that do not require VPA uptake in the brain. The results suggest that increasing plasma LPCs, e.g., via stimulated secretion of LPCs from the liver might be useful to protect the brain after injury. The potential risk of steatohepatosis must be carefully considered for prolonged treatment. The liver toxicity of VPA is both time and dose dependent and thought to arise from 4-ene-VPA metabolites that enter the mitochondria for detoxification via beta-oxidation, leading to a depletion of coenzyme-A, glutathione and acylcarnitines [83,84]. In this context, it is important to note that plasma levels of long-chain acylcarnitines were increased in CCI mice as compared to sham mice both in vehicle and VPA treatment groups, which is in agreement with previous studies in TBI mice [85] and TBI patients [86]. In our settings, VPA neither aggravated nor attenuated the increase in this marker.

It is a further limitation of our study that we have not monitored liver function tests during VPA treatment and that the observation and treatment periods were short, and biological correlates were only obtained at one time point 7 days after trauma, which is a short survival time only encompassing the acute to subacute phase of TBI. Longer survival times are needed to address the putative beneficial effects of VPA in supporting the long-term rehabilitation process. On the other hand, shorter survival times would be required to monitor in more detail the onset of beneficial or adverse pharmacological effects of VPA. Furthermore, we did not address putative sex differences in the present study and used only adult male C57BL/6N mice, because the introduction of sex as an additional variable would have required a doubling of animal numbers. There are sex differences to susceptibility to prenatal VPA as a model of autism, which however do not apply to VPA as anticonvulsant in adult mice. In humans, substantial sex differences in lipidomic and metabolomic plasma profiles have been described and arise in part from contraceptives [87] but are not well studied in mice, and putative sex differences need to be addressed in future studies.

## 5. Conclusions

We show that post-injury VPA treatment in a murine model of TBI reduces hippocampal damage, microglial activation and pro-inflammatory gene expression, which were largely independent of VPA’s HDACi functions and, in particular, independent of an REST/NRSF mediated de-repression of target gene expression. Instead, neuroprotection and anti-neuroinflammation appear to arise at least in part from the beneficial peripheral changes of lipidomic patterns that manifest in increased plasma LPCs and hence lead to the higher brain availability of LPCs. It may be speculated that VPA stimulates the secretion of LPCs from the liver, which is beneficial for the injured brain and appears to be safe for the mouse liver at least for short-term treatment. However, considering VPA PK differences between mice and humans, one cannot predict the outcome in human TBI, and future studies are warranted to understand how the modulation of cellular acetylation affects the interplay between gene and protein regulation, lipid metabolism and brain-to-liver communication.

## Figures and Tables

**Figure 1 cells-13-00734-f001:**
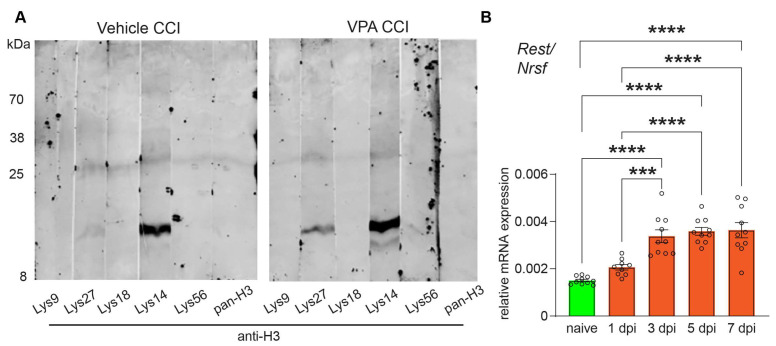
VPA functions as HDACi and REST/NRSF gene expression is up-regulated after CCI. (**A**) Immunoblot showing the pharmacologic effect of valproic acid (VPA) with increased acetyl-histone H3 levels in perilesional brain samples of VPA-treated mice (pooled from *n* = 11 per group) Antibodies used: 1: acetyl-histone H3 (Lys9) (C5B11); 2: acetyl-histone H3 (Lys27) (D5E4); 3: acetyl-histone H3 (Lys18) (D8Z5H); 4: acetyl-histone H3 (Lys14) (D4B9); 5: acetyl-histone H3 (Lys56); 6: histone H3 (D1H2). (**B**) Gene expression analyses and quantification of the transcription factor REST/NRSF normalized to Ppia in naïve animals (*n* = 10) and at 1 (*n* = 9 animals), 3, 5, and 7 (each time point *n* = 10 animals) days after CCI. Data are expressed as mean ± SEM with individual values shown and *p* values were calculated by one-way ANOVA with post hoc Holm–Šidák correction (***
*p* < 0.001; **** *p* < 0.0001).

**Figure 2 cells-13-00734-f002:**
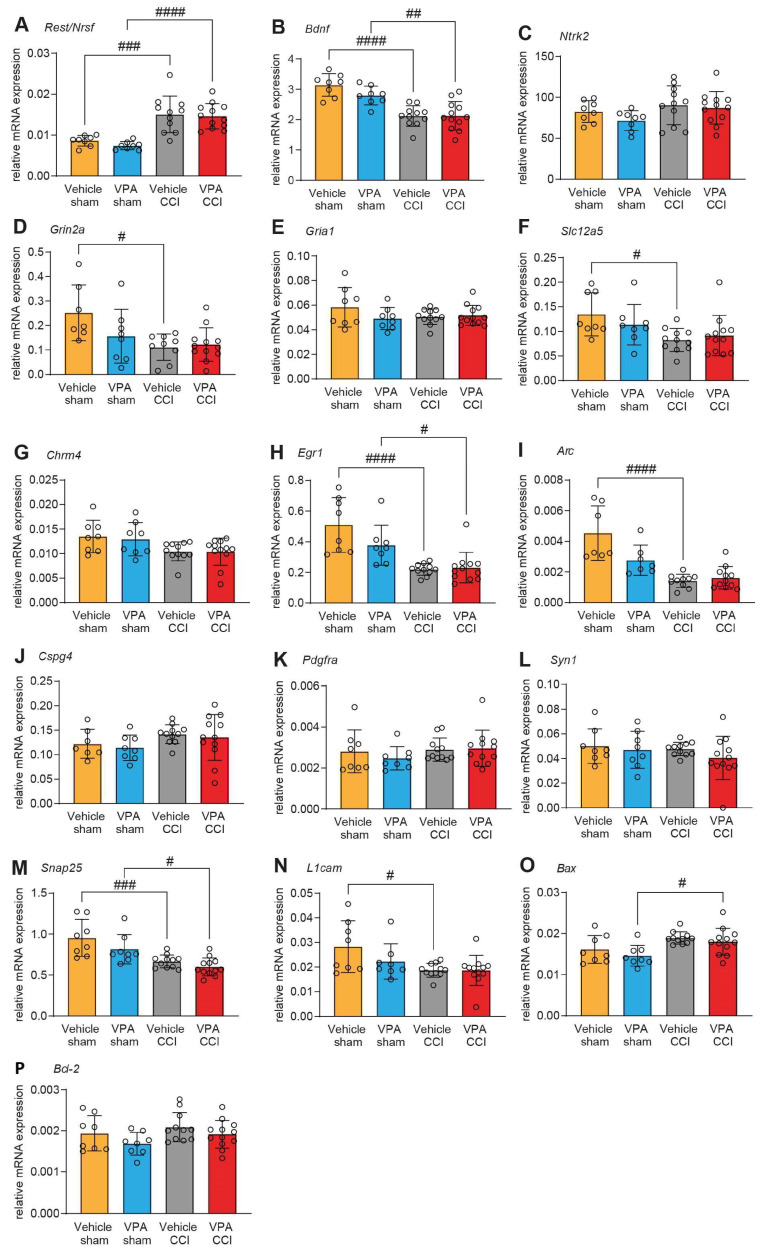
VPA treatment does not affect mRNA expression of REST/NRSF target genes in the injured brain. Gene expression analyses by qPCR 7 days post-injury (dpi) relative to *Ppia*. (**A**) *Rest*/*Nrsf*; (**B**) *Bdnf*; (**C**) *Ntrk2*; (**D**) *Grin2a*; (**E**) *Gria1*; (**F**) *Slc12a5*; (**G**) *Chrm4*; (**H**) *Egr1*; (**I**) *Arc*; (**J**) *Cspg4*; (**K**) *Pdgfra*; (**L**) *Syn1*; (**M**) *Snap25*; (**N**) *L1cam*; (**O**) *Bax*; (**P**) *Bcl2*. CCI effects were observed, but VPA treatment had no effect on gene expression in sham or CCI groups. Data are expressed as mean ± SEM with individual values shown and *p* values were calculated by one-way ANOVA with post hoc Holm–Šidák correction (# *p* < 0.05; ## *p* < 0.01; ### *p* < 0.001; #### *p* < 0.0001).

**Figure 3 cells-13-00734-f003:**
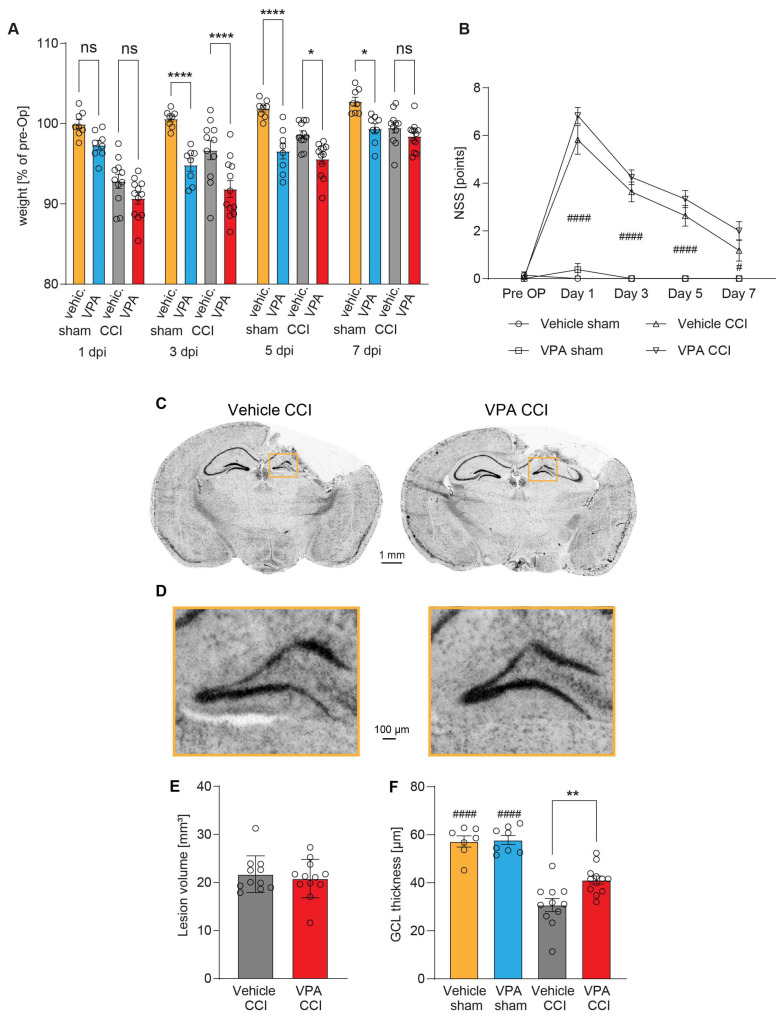
Administration of VPA does not influence acute neurological deficits or brain lesion size at 7 dpi but attenuates structural damage in the hippocampus. (**A**) Body weight time course at post-traumatic day 1, 3, 5 and 7 in % of pre-surgery body weight [g], ns = not significantly different. (**B**) Neurological severity score (NSS) day 1, 3, 5 and 7. Sample size NSS/body weight: vehicle-controlled cortical impact (CCI): *n* = 11, VPA CCI: *n* = 12, vehicle sham: *n* = 8, VPA sham: *n* = 8. * *p* < 0.05, **** *p* < 0.0001 significantly different as indicated (CCI vs. sham animals). (**C**) Representative images of cresyl violet stained brain sections at 7 dpi from vehicle or VPA-treated mice. (**D**) Boxed regions from (**C**) shown in higher magnification with detail enlargement of the hippocampal granule cell layer (GCL) at 7 dpi in vehicle and VPA-treated mice. (**E**) Quantification of lesion volume and (**F**) GCL thickness (vehicle CCI: *n* = 11, VPA CCI: *n* = 12), * indicates significance levels between CCI groups and # between sham and corresponding CCI groups (** *p* < 0.01, #### *p* < 0.0001). All data points represent individual animals, and data are expressed as mean ± SEM, *p* values were calculated by two-way ANOVA with Holm–Šidák correction (**A**), Kruskal–Wallis test with Dunn’s correction (**B**), Mann–Whitney U Test (**E**) and one-way ANOVA with Holm–Šidák correction (**F**).

**Figure 4 cells-13-00734-f004:**
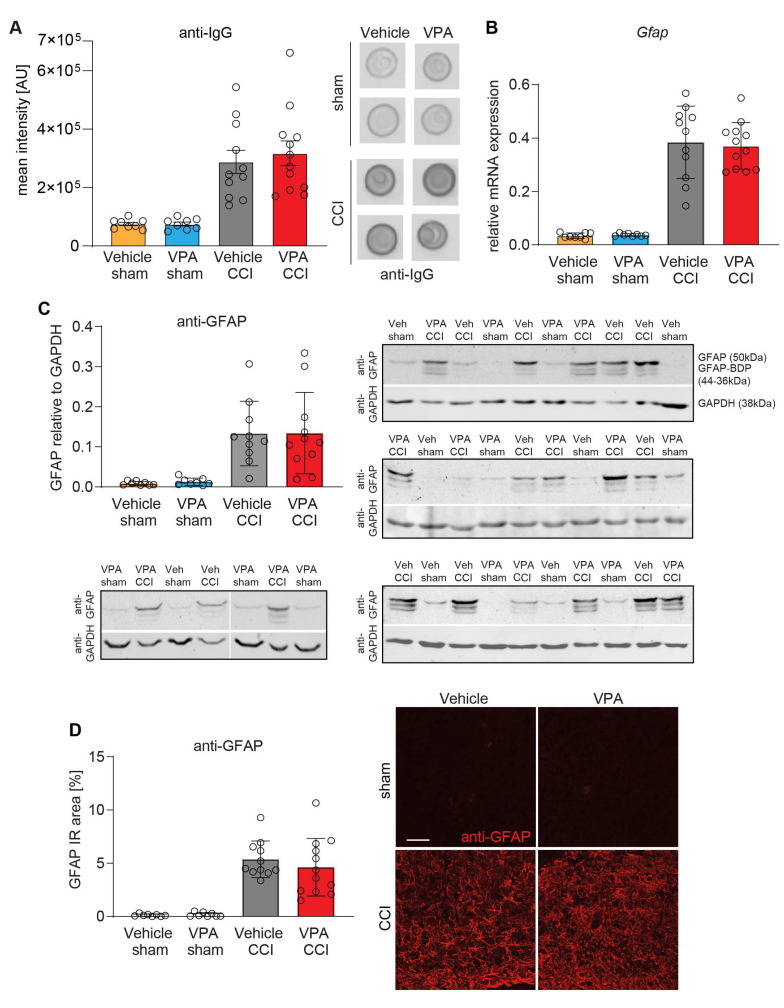
VPA treatment does not affect BBB disruption and astrocyte activation at 7 dpi. (**A**) Quantification of IgG extravasation into brain tissue. Representative examples of immuno-dot blots from the lesioned hemisphere at 7 dpi and corresponding sham tissue are shown. (**B**) Gene expression analysis of the astrogliosis marker *GFAP* normalized to *Ppia*. (**C**) Quantification of GFAP protein levels and breakdown products (BDPs) relative to GAPDH. One sample from vehicle CCI and VPA CCI was excluded because the corresponding GAPDH reference band could not be correctly quantified. Anti-GFAP/anti-GAPDH immunoblots are shown. (**D**) Quantification of the anti-GFAP immunoreactivity (IR) area. Representative examples of immunofluorescence images are shown (scale: 50 µm). Data are expressed as mean ± SEM with individual values shown, *p* values were calculated by one-way ANOVA followed by comparison of pre-defined pairs (Vehicle versus VPA treatment) using *t*-test with α adjustment according to Holm–Šidák.

**Figure 5 cells-13-00734-f005:**
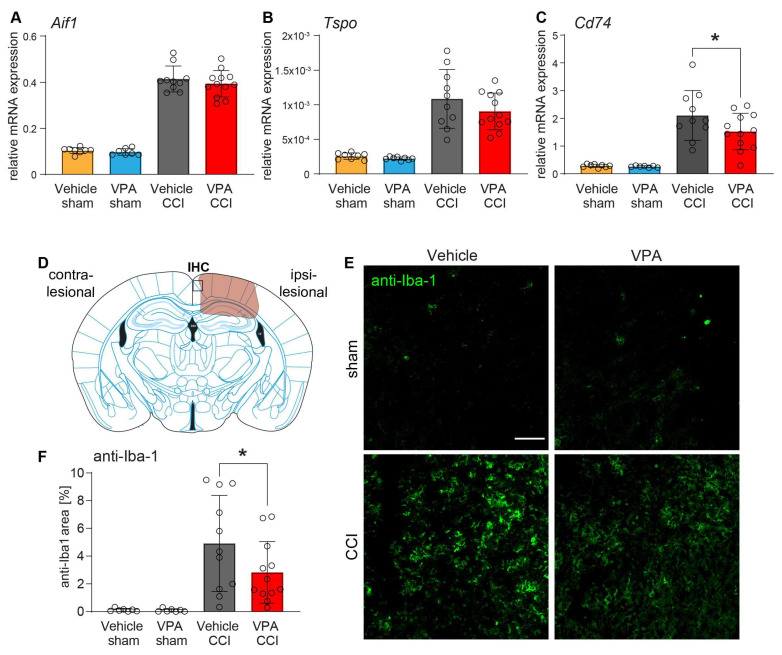
Microglia activation after CCI is reduced by VPA. Gene expression analyses of the microglia markers *Aif1* (**A**), *Tspo* (**B**, 18 kDa translocator protein) and *Cd74* (**C**) normalized to *Ppia*. (**D**) Scheme showing the lesion core (orange) and the region of interest for the examination of microglial activation by anti-Iba1 immunohistochemistry (IHC, black box) medial to the lesion site. (**E**) Images of anti-Iba1 immunostaining showing perilesional microglia activation 7 dpi (scale 50 µm). (**F**) Quantification of the anti-Iba-1 immunoreactive area medial to the lesion site. Sample size: vehicle CCI: *n* = 11, VPA CCI: *n* = 12, vehicle sham: *n* = 8, VPA sham: *n* = 8. Data are expressed as mean ± SEM with individual values shown and *p* values (* *p* < 0.05) were calculated by one-way ANOVA followed by comparison of the treatment-relevant groups (CCI-Vehicle versus CC-VPA) using *t*-test.

**Figure 6 cells-13-00734-f006:**
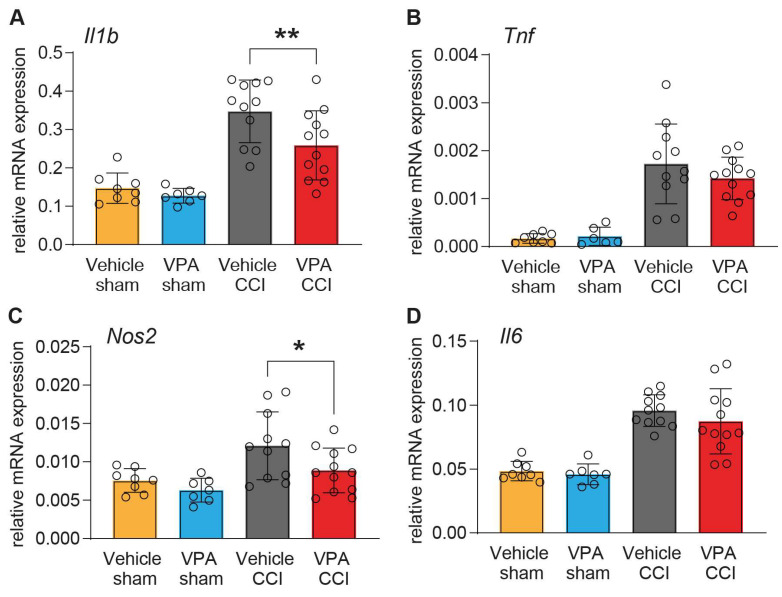
VPA attenuates pro-inflammatory gene expression after CCI. Gene expression analyses of pro-inflammatory markers normalized to *Ppia*. Quantification of the pro-inflammatory cytokines *Il1b* (**A**), *Tnfa* (**B**), *Nos2* (**C**) and *Il6* (**D**). Sample size vehicle CCI: *n* = 11, VPA CCI: *n* = 12. Outliers as determined by Grubb’s test were removed from further analysis (*Il1b*, *Nos2*, *Il6*: 1 outlier in VPA sham, *Tnfa*: 2 outliers in VPA sham). Data are expressed as mean ± SEM with individual values shown and *p* values (* *p* < 0.05, ** *p* < 0.01) were calculated by one-way ANOVA followed by comparison of pre-defined pairs (Vehicle versus VPA treatment) using *t*-test with α adjustment according to Holm–Šidák.

**Figure 7 cells-13-00734-f007:**
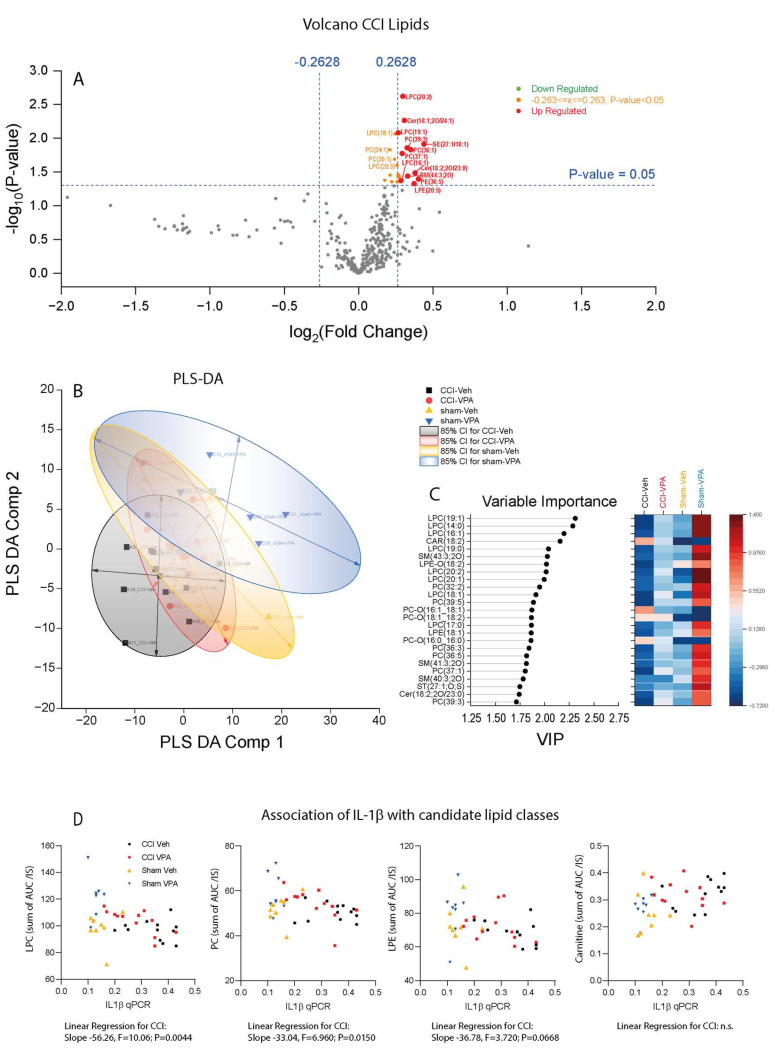
Lipidomic analysis of plasma samples 7 days after controlled cortical impact (CCI) or sham surgery in animals treated with vehicle or VPA (3 dpi 400 mg/kg VPA i.p.). (**A**) Volcano plot of CCI-Vehicle versus CCI-VPA. Lipids increased in the VPA group are on the right side of the x-axis (sham groups in Appendix A). (**B**) Partial least square discriminant analysis (PLS-DA) component 1 versus component 2 scatter plot and 85% confidence ellipses. Two sham-VPA animals were excluded, which were identified as outliers by random forest analysis. (**C**) Variable importance plot (VIP) of PLS-DA component-1. (**D**) XY-scatter plots showing the association of lipid classes versus *Il1b* mRNA. To obtain a summary value of a lipid class, individual AUC/IS values of individual lipid species with different chain lengths and saturation of the respective class were summed. Associations with TNFα are shown in Appendix A. Sample sizes: controlled cortical impact (CCI) vehicle *n* = 12, CCI VPA *n* = 12, sham vehicle *n* = 8, sham-VPA *n* = 8. Abbreviations of lipids: CAR, carnitines; CER, ceramides; LPC, lysophosphatidylcholine; PC, phosphatidylcholine; LPE, lysophosphatidylethanolamine; SE, steryl ester; SM, sphingomyelins; ST, sterols.

**Figure 8 cells-13-00734-f008:**
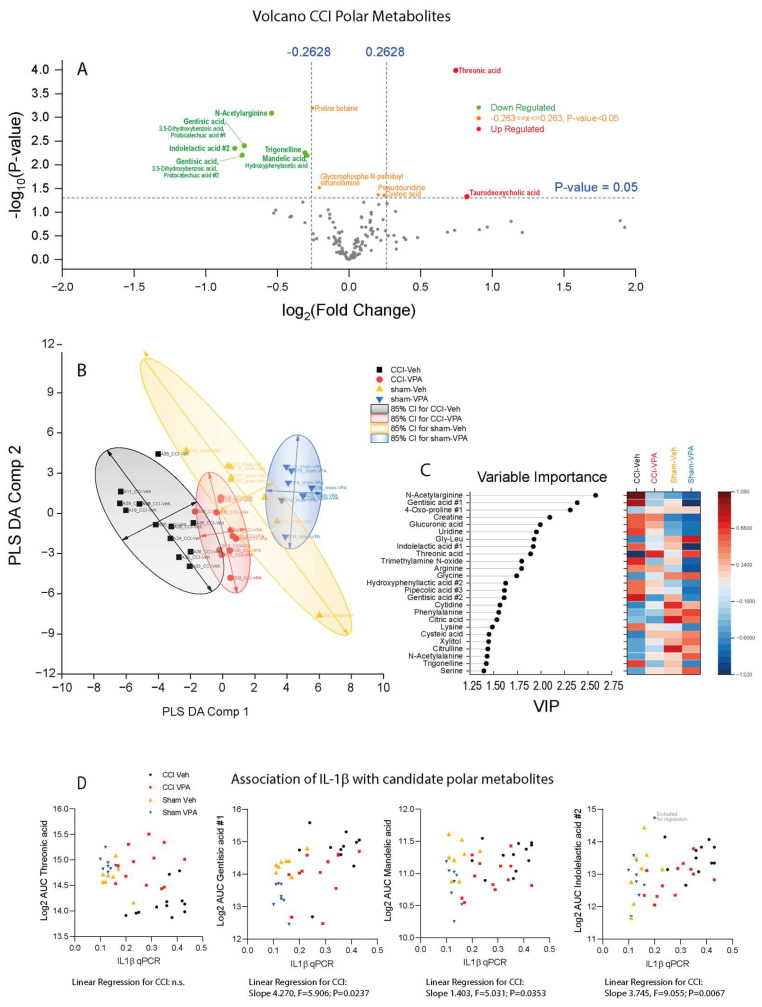
Metabolomic analysis of polar metabolites in plasma samples 7 days after controlled cortical impact (CCI) or sham surgery in animals treated with vehicle or VPA (3 dpi 400 mg/kg VPA i.p.). (**A**) Volcano plot of CCI-Vehicle versus CCI-VPA. Metabolites that were increased in the VPA group are on the right side of the x-axis (sham groups in Appendix A). (**B**) Partial least square discriminant analysis (PLS-DA) component 1 versus component 2 scatter plot and 85% confidence ellipses. One VPA animal each in the CCI and sham group were excluded, which were identified as outliers by random forest analysis. (**C**) Variable importance plot (VIP) of PLS-DA component-1. (**D**) XY scatter plots showing the association of key regulated metabolites versus *Il1b*. Associations with *Tnfa* are shown in Appendix A. Sample sizes: controlled cortical impact (CCI) vehicle *n* = 12, CCI-VPA *n* = 12, sham vehicle *n* = 8, sham-VPA *n* = 8.

**Table 1 cells-13-00734-t001:** REST/NRSF target genes.

Biological Processes	Neurotrophic Factors	Ion Channel/ Transporter Activity	Transcription Factors	Oligodendrocyte Differentiation	Synapse Plasticity /Organization	Apoptosis
GO terms	GO:0031547	GO:0015075 GO:0005216 GO:0006811	GO:0003700	GO:0048709 GO:0048008	GO:0051823 GO:0050808	(GO:0097190) GO:0006915
target genes	Bdnf Ntrk2	Glutamate: Grin2a Gria1 Chloride: Slc12a5 Acetylcholine: Chrm4	Egr1 Arc	Cspg4 Pdgfra	Syn1 Snap25 L1cam	Bax Bcl-2

## Data Availability

The datasets generated and analyzed during the current study are included in this published article and its Supporting Information. Further raw data are available from the authors upon reasonable request.

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
