# Peer review of "Valproic Acid Treatment after Traumatic Brain Injury in Mice Alleviates Neuronal Death and Inflammation in Association with Increased Plasma Lysophosphatidylcholines"

_cells, 2024, doi:10.3390/cells13090734_

Round 1
Reviewer 1 Report
Comments and Suggestions for Authors
The topic is significant and the obtained results are original and provide new information on the effects of VPA following TBI. The experimental procedure is well-designed and the initial hypotheses are well-stated and demonstrated.
In my opinion, the work can be published after some corrections.
Here there are some comments and requests.
Typos
Line 194: Please, correct the punctuations.
Line 241, “Supplemental Figure 1”: It should be Supplemental Figure 2. Please, correct.
Line 393, “..up-regulation of GFAP both at the mRNA (Fig. 4C)...” It should be Figure 4B. Please correct
Lines 293-294, “…and protein level (Fig. 4B, D)…” It should be Fig.s 4C, D. Please correct.
Line 417, “compared to vehicle treatment (Fig. 5D)” It should be Fig. 5F. Please correct.
Line 473: Please explain the meaning of the acronym LPE that was first used in this line.
Line 469 and line 492, “TNFα (Suppl. Fig. 6)” Please, specify that it is Suppl Fig 6A.
Lines 486-487, “(sham 486 groups in Suppl. Fig. 3)” It should be Suppl. Fig 4A. Please correct.
Line 520, “…Tnfa are shown in Suppl. Fig. 6…” Please, specify that it is Suppl Fig 6B.
Experimental analysis
Line 97, “and received VPA or vehicle 15 min…”: I suggest excluding this time point. It is not further considered in the performed experiments and analysis and in the reported results.
Lines 302-305, “...no indications of severe side effects such as hepatotoxicity, pancreatic dysfunction or coagulation abnormalities were observed”: Have you run any tests or collected any observations to justify this statement? Please clarify.
Line 404: “Representative anti-GFAP immunoblots are shown”. As you provide all the immune dot blot images for IgG quantification, you should provide all the immunoblot images for GFAP quantification.
Line 405: “Representative examples of immunofluorescence images are shown” You should provide all the immunofluorescence images for GFAP quantification.
Lines 415-416, “We assessed microglia via anti-Iba1 immunohistochemistry in the perilesional cortex” It is not an immunohistochemistry, but an immunofluorescence since the detection system is based on fluorochrome-conjugated secondary antibodies. Please correct this in the Materials and Methods section too (line 208).
Lines 416-417, “Iba1 immunoreactivity was attenuated in VPA-treated mice as compared to vehicle treatment (Fig. 5D).” In which group? CCI mice or sham mice? Please clarify.
Lines 424-425, “(E) Images of anti-Iba1 immunostaining showing perilesional microglia activation 7 dpi”. You should provide all the immunofluorescence images for Iba1 quantification.
Lines 508-510: You should add some information regarding the relationships between key metabolites and TNFα. Moreover, it would be best to refer to the corresponding figures (Fig.8D and Suppl. Fig.6B) for clarity.
Author Response
Response to Reviewer 1:
Typos
Line 194: Please, correct the punctuations.
Line 241, “Supplemental Figure 1”: It should be Supplemental Figure 2. Please, correct.
Line 393, “..up-regulation of GFAP both at the mRNA (Fig. 4C)...” It should be Figure 4B. Please correct
Lines 293-294, “…and protein level (Fig. 4B, D)…” It should be Fig.s 4C, D. Please correct.
Line 417, “compared to vehicle treatment (Fig. 5D)” It should be Fig. 5F. Please correct.
Line 473: Please explain the meaning of the acronym LPE that was first used in this line.
Line 469 and line 492, “TNFα (Suppl. Fig. 6)” Please, specify that it is Suppl Fig 6A.
Lines 486-487, “(sham 486 groups in Suppl. Fig. 3)” It should be Suppl. Fig 4A. Please correct.
Line 520, “…Tnfa are shown in Suppl. Fig. 6…” Please, specify that it is Suppl Fig 6B.
Our response: We thank you for your careful review. We have corrected the items listed above accordingly and rechecked figures assignments throughout. The changes are highlighted in red writing.
Experimental analysis
Line 97, “and received VPA or vehicle 15 min…”: I suggest excluding this time point. It is not further considered in the performed experiments and analysis and in the reported results.
Our response: The sentence in question describes the treatment regimen and we consider the complete information necessary even if no experiments or analyzes have been carried out at this time-point.
Lines 302-305, “...no indications of severe side effects such as hepatotoxicity, pancreatic dysfunction or coagulation abnormalities were observed”: Have you run any tests or collected any observations to justify this statement? Please clarify.
Our response: The absence of severe side effects on hepatotoxicity, pancreatic dysfunction, or coagulation abnormalities was concluded based on the lack of obvious macroscopic effects, and based on plasma lipidomic and metabolomic analyses, which did not reveal severe organ dysfunctions. We agree that this statement should be substantiated by a more detailed analysis. Unfortunately, we did not made images of organs or collected biopsies for further analysis. We have therefore now omitted this statement.
Line 404: “Representative anti-GFAP immunoblots are shown”. As you provide all the immune dot blot images for IgG quantification, you should provide all the immunoblot images for GFAP quantification.
Our response: All immunoblot images including anti-IgG, anti-GFAP, and anti-GAPDH have been provided in the file “original blots” on page 1 (IgG) and page 2 (GFAP, GAPDH).
Line 405: “Representative examples of immunofluorescence images are shown” You should provide all the immunofluorescence images for GFAP quantification.
Lines 424-425, “(E) Images of anti-Iba1 immunostaining showing perilesional microglia activation 7 dpi”. You should provide all the immunofluorescence images for Iba1 quantification.
Our response: More than 200 images were acquired for the quantification of anti-GFAP/anti-Iba1 immunofluorescence. We now provide a selection of representative images from the ipsilesional hemisphere of CCI-Vehicle and CCI-VPA mice (n=10/group), which is presented as Supplemental Figure S3B. The reference to this supplementary panel was added to the text. We are confident that these images demonstrate the quantified differences and that showing all acquired images from sham and CCI mice, ipsi- and contralesional images taken at two different Bregma levels is not appropriate.
Lines 415-416, “We assessed microglia via anti-Iba1 immunohistochemistry in the perilesional cortex” It is not an immunohistochemistry, but an immunofluorescence since the detection system is based on fluorochrome-conjugated secondary antibodies. Please correct this in the Materials and Methods section too (line 208).
Our response: We now write “immunofluorescence analyses “ or “fluorescent immunohistochemistry” as to our understanding immunohistochemistry is defined as the selective identification of an antigen in tissues by antigen specific-antibodies, independent of the detection system.
Lines 416-417, “Iba1 immunoreactivity was attenuated in VPA-treated mice as compared to vehicle treatment (Fig. 5D).” In which group? CCI mice or sham mice? Please clarify.
Our response: We now write: “Iba1 immunoreactivity was attenuated in VPA-treated CCI mice as compared to vehicle-treated CCI mice”.
Lines 508-510: You should add some information regarding the relationships between key metabolites and TNFα. Moreover, it would be best to refer to the corresponding figures (Fig.8D and Suppl. Fig.6B) for clarity.
Our response: We have highlighted the reference to the respective figures in the text which might have been overlooked. The relationship of the key metabolites with cytokines is presently not known. This is the first paper showing a negative correlation of the key metabolites (LPC and threonic acid) with Il1b and Tnfa in the context of traumatic brain injury. It has been shown previously that treatment with magnesium threonate reduces pain in a model of neuropathic pain in mice by inhibiting NFkappaB and thereby production of TNFalpha (PMID: 28306698).
Reviewer 2 Report
Comments and Suggestions for Authors
The manuscript by Hummel and co-workers describes a mouse study investigating the neuroprotective and anti-inflammatory effects of valproic acid (VPA) systemic administration in experimental traumatic brain injury (TBI). The manuscript is well and clearly written, the experimental procedures are appropriate as well as the data analysis and discussion of the results. I have only minor comments in order to improve the quality of the manuscript.
- in the text (page 7, lines 302-305) it is stated that no indication of side effects was found following VPA treatment, such as body weight loss, hepatotoxicity, pancreatic dysfunction or coagulation abnormalities. It would be useful to provide these data in a Supplementary table.
-Page 2, line 57 : « animals models » should be replaced by « animal models ».
-It would be important to justify the choice of the sex and strain of the subjects.
-The use of male mice only should be mentioned as a possible limitation of the study in the dedicated section of the discussion.
Author Response
Response to Reviewer 2
The manuscript by Hummel and co-workers describes a mouse study investigating the neuroprotective and anti-inflammatory effects of valproic acid (VPA) systemic administration in experimental traumatic brain injury (TBI). The manuscript is well and clearly written, the experimental procedures are appropriate as well as the data analysis and discussion of the results. I have only minor comments in order to improve the quality of the manuscript.
- in the text (page 7, lines 302-305) it is stated that no indication of side effects was found following VPA treatment, such as body weight loss, hepatotoxicity, pancreatic dysfunction or coagulation abnormalities. It would be useful to provide these data in a Supplementary table.
Our response: The absence of severe side effects on hepatotoxicity, pancreatic dysfunction, or coagulation abnormalities was concluded based on the lack of obvious macroscopic effects, and by plasma lipidomic and metabolomic analyses, which did not reveal serious organ dysfunctions. We agree that this statement should be substantiated by a more detailed analysis. Unfortunately, we did not made images of organs or collected biopsies for further analysis. We have therefore now omitted this statement.
-Page 2, line 57 : « animals models » should be replaced by « animal models ».
Our response: done
-It would be important to justify the choice of the sex and strain of the subjects.
-The use of male mice only should be mentioned as a possible limitation of the study in the dedicated section of the discussion.
Our response: We now write in the methods (2.1. Mice): Male C57BL6/N mice (Janvier, ~22 g and 6 weeks old), a commonly used mouse strain in experimental research, were housed individually and maintained in a controlled environment (12-hour dark/light cycle, 23°C, 55 % humidity) with food and water ad libitum. Only male mice were included to exclude fluctuations in the estrus-dependent hormonal status as possible confounding factors.
We also now explain that the use of male mice only is a limitation of our study (Line 648 pp). “Sex” as additional variable would have required doubling of animal numbers. Lipidomic and metabolomic plasma profiles in humans depend on sex and age, and VPA pharmacokinetics also may depend on sex. Hence, for the present study it would have been too complex to include male and female mice with/without CCI, with/without VPA.
Reviewer 3 Report
Comments and Suggestions for Authors
The paper entitled “Valproic acid treatment after traumatic brain injury in mice alleviates neuronal death and inflammation in association with increased plasma Lyphosphatidylcholines by Regina Hummel and the co-authors is a nice contribution to highlighting the neuroprotective mechanism of valproic acid against traumatic brain injury-induced neuroinflammation and neurodegeneration. According to the overall findings “TBI-upregulated the expression of REST/NRSF gene. Moreover, with the administration of valproic acid, neurological deficits, brain lesion size, blood-brain barrier permeability, or astrogliosis were not affected and REST/NRSF target genes were only marginally influenced by VPA. However, VPA attenuated structural damage in the hippocampus, microgliosis, and expression of the pro-inflammatory marker genes. Indeed, repurposing is the best approach to accelerate the drug discovery processes and mitigate these devastating health hazards.
1. All abbreviations in the abstract may be defined, it will be helpful for search and a quick overview of the abstract.
2. The paper has been nicely presented with clear graphs and statistical analysis.
3. A very clear research gap or limitation has been given in the paper which is a plus.
4. What was the rationale for using Male Mice? Is the disease associated with males only? How many mice were taken and given a reason or scientific calculation?
5. Collectively, the study has been nicely presented, which may be processed for publication.
Comments on the Quality of English Language
English language required minor editing
Author Response
Response to Reviewer 3
- All abbreviations in the abstract may be defined, it will be helpful for search and a quick overview of the abstract.
Our response: We are grateful for this suggestion and defined the abbreviations REST/NRSF, TBI, Il1b, and Tnf.
- The paper has been nicely presented with clear graphs and statistical analysis.
- A very clear research gap or limitation has been given in the paper which is a plus.
- What was the rationale for using Male Mice? Is the disease associated with males only? How many mice were taken and given a reason or scientific calculation?
Our response: Only male mice were included to exclude fluctuations in the estrus-dependent hormonal status as possible confounding factors. We added this information in the methods section (2.1. Mice). We also now explain that the use of male mice only is a limitation of our study (Line 648 pp). “Sex” as additional variable would have required doubling of animal numbers. Lipidomic and metabolomic plasma profiles in humans depend on sex and age, and VPA pharmacokinetics also may depend on sex. Hence, for the present study it would have been too complex to include male and female mice with/without CCI, with/without VPA.
The number of mice investigated in this study was based on power analysis as described in the methods (2.1. Mice). “Animal numbers were determined by power analysis using experience from prior experiments (37). The sample size of the CCI groups was based on the assumption that a 20% change in brain lesion size as the main outcome parameter is relevant. The probability of type-1 error was set to α = 0.05, and type-2 error was set at β = 0.2".
- Collectively, the study has been nicely presented, which may be processed for publication.
Our response: We are grateful for the positive comments on the presentation of our results.